# Insight in the development of the mutual trust relationship between trainers and trainees in a workplace-based postgraduate medical training programme: a focus group study among trainers and trainees of the Dutch general practice training programme

Linda H.A. Bonnie ![ORCID],[1] Mechteld R.M. Visser,[1] Anneke W.M. Kramer,[2] Nynke van Dijk[1]

For numbered affiliations see end of article.

**Correspondence to**
Linda H.A. Bonnie;
l.h.bonnie@amsterdamumc.nl

## ABSTRACT

**Objectives** Trust plays an important role in workplace-based postgraduate medical education programmes. Trainers must trust their trainees for granting them greater independence. Trainees must trust their trainer for a safe learning environment. As trainers' and trainees' trust in each other plays an important role in trainee learning and development, the authors aimed to explore the development of the mutual trust relationship between trainers and trainees.

**Setting** This study was performed in a general practice training department in the Netherlands.

**Participants** All trainers and trainees of the general practice training department were invited to participate. Fifteen trainers and 34 trainees, voluntarily participated in focus group discussions.

**Outcome measures** The authors aimed to gain insight in the factors involved in the development of the mutual trust relationship between trainers and trainees, in order to be able to create a model for the development of a mutual trust relationship between trainers and trainees. The risk-based view of trust was adopted as leading conceptual framework.

**Results** In the first stage of trust development, trainers and trainees develop basic trust in each other. Basic trust forms the foundation of the trust relationship. In the second stage, trainers develop trust in trainees taking into account trainees' working and learning performance, and the context in which the work is performed. Trainees trust their trainer based on the trainer'savailability and accessibility and the personal relationship between the trainee and their trainer. Trainee self-confidence modifies the development of a trust relationship.

**Conclusion** The development of a mutual trust relationship between trainers and trainees is a complex process that involves various stages, goals, factors and interactive aspects. As the mutual trust relationship influences the learning environment for trainees, greater emphasis on the mutual trust relationship may improve

## Strengths and limitations of this study

► We used the risk-based view of trust as the leading conceptual framework for this study.
► We included a relatively large amount of trainers and trainees in various stages of the training programme, which provided us with many different experiences and view points.
► This study is a single-centre study, which may cause results to be not directly applicable in other postgraduate medical education programmes.
► Trainers included in the study were only involved in long educational relationships, so their vision on the influence of the duration of the educational relationship on mutual trust development could not be explored.

learning outcomes. Further research may explore the effect of long-term and short-term educational relationships on the trust relationship between trainers and trainees.

## INTRODUCTION

Trust (*noun*): 'Firm belief in the reliability, truth, or ability of someone or something'.[1]

Entrust (*verb*): 'To put one's trust in a person, with regard to a particular task or responsibility'.[2]

Trust is a complex feeling, not easy to summarise in a simple definition. The amount of trust someone has in another person is highly dependent on personal factors, such as thoughts and motivations, as well as on the context in which the trust has

to be awarded.[3–5] Additionally, trust is not an established fact, as it can be gradual and variable.[5]

Trust is fundamental to workplace-based learning in postgraduate medical education (PGME) programmes.[3] In workplace-based PGME programmes, trainees learn to work independently when providing patient care, with the goal of becoming independently functioning medical specialists.[6 7] In the process of learning to work independently, trainees are supervised by trainers, who are educators for trainees, next to their work as patient care providers.[8] In the process of supervision, trainers must balance the freedom they give their trainees in independently providing patient care, against patient safety. Trainers thus have to trust their trainee to provide good quality of care for their patients.[3 9 10] In this process of trust development, trainers take into account factors that are related to themselves as a trainer and as a patient care provider, the trainee, the trainer–trainee relationship, the context in which the work is performed and the nature of the task in question.[3 10–15] Once trainers trust their trainee, they can gradually entrust them with performing patient care independently.[3]

Trust is not just important in terms of patient safety, it is also a prerequisite for successful trainee learning. Being trusted by a trainer allows trainees to increasingly learn to work independently in the workplace and thereby develop themselves towards becoming independently functioning medical specialists.[3] Additionally, trainers entrusting trainees to perform tasks independently can boost motivation for learning and working.[3 11] Also, for trainees being able to learn from experiences in the workplace, to perform their work properly and being able to receive feedback, a safe learning environment is required. Trusting their trainer is essential for trainees to experience a safe learning environment.[16–20] However, factors involved in the process of trainees developing trust in their trainer still are poorly understood.

As both trainer's trust in their trainee,[3] as well as trainee's trust in their trainer,[16–20] play an important role in trainee learning and development, understanding the mutual trust relationship might help to optimise the learning outcomes for trainees.[16 17] As factors involved in the process of trust development of trainees in their trainer still remain poorly understood, and little information is available about how the trust development processes of both trainers and trainees may influence each other, we aim to gain insight in the development of the mutual trust relationship between trainers and trainees in a workplace-based PGME programme.

## METHODS

### Context

This focus group study was conducted among the trainers and trainees of a Dutch GP-training programme. Dutch GPs are a pivotal element of the Dutch healthcare system. In the event of a medical problem (life-threatening situations excluded), all Dutch in habitants first visit their general practitioner (GP) for advice and treatment. If necessary the GP refers to a medical specialist, for further evaluation.[21 22] To prepare trainee GPs for this broad field of work, the Dutch GP-training programme involves a significant amount of workplace-based learning. The first and third year of the training programme are spent working in general practice. During this period, trainees are supervised by one or two specially trained GPs, who coach, teach and assess the trainee. The second year of their training programme is dedicated to 3–6 months traineeships in the areas of clinical or emergency medicine, care for the elderly and psychiatry. The trainers in these internships are medical specialists who have been trained to supervise trainee GPs.[23 24]

### Study design

As we aimed to explore the process of development of the mutual trust relationship between trainers and trainees in a workplace-based PGME programme, we adopted a qualitative approach. As leading conceptual framework, we chose to adopt a risk-based view of trust by Das and Teng.[25] By using this framework, we assume that trust and risk are theoretically each other's opposites. This means that when we trust someone, we experience a low risk in trusting the other person (perceived risk), and we are willing to accept that risk. On the other hand, when we do not trust someone, the perceived risk in trusting the other person is high, and we are not willing to accept that risk. The feeling of a certain amount of trust in someone is called subjective trust, the perception of trust. This subjective trust is influenced by personal and situational factors, the so-called trust antecedents in the terminology of Das and Teng.[25] The interplay between subjective trust and trust antecedents leads to certain behaviour in taking risks and trusting someone, also known as behavioural trust.[25]

Within the framework of a risk-based view of trust, the consequences of behavioural trust may again inform and influence a person's subjective trust. Additionally, Das and Teng believe that trusting somebody, and therefore taking a risk in that person, can boost the development of a mutual trust relationship.[25] We use these assumptions as the basis of our study, as we aim to explore how the mutual trust relationship between trainers and trainees develops, and which factors are involved in this process.

We chose to use focus group discussion with trainers and trainees for data collection, which we believed would provide insight in participants' experiences and views regarding the development of trust. Individual differences in experience and views could be shared and explored in interactive group discussions.[26 27]

For the analysis of the data, we used thematic analysis, as this is a method for 'identifying, analysing and reporting patterns within data'.[28] With the thematic analysis, we aimed to gain insight in the process of development of the mutual trust relationship between trainers and trainees. Although an extensive amount of literature is available about trainers developing trust in their trainees,[3 10–15] we

chose to apply an inductive approach for the coding and analysis, as we hoped to identify factors involved in the development of the mutual trust relationship between trainers and trainees that currently are poorly understood an not yet described in the literature.[28 29]

## Participants

We conducted this study among GP trainers and trainee GPs from the Amsterdam University Medical Centers (UMC)—location Academic Medical Center (AMC) GP training institute. We selected homogeneous groups, that is, separate groups for trainers and trainees, as we expected that discussion of some aspects of trust could place participants in a vulnerable position, and we hoped that eliminating hierarchy in the focus groups would promote free and open discussions.[26]

All trainers and trainees from the Amsterdam UMC—location AMC GP training institute were informed about the purpose of the study, in person and by email, by the principle researcher (LB). Focus group meetings for trainers were held during the faculty development programme. Trainers could voluntary sign up for participation in the focus group discussions, after having received information about the study. Trainees participated in focus groups during their weekly educational day, they could voluntarily sign up for participation in the focus group discussions after information about the study was provided to them.

## Data collection

The data used in this study were obtained from a larger study into the implementation of Entrustable Professional Activities (EPAs) in the GP-training programme. The practical aspects of the use of EPAs in the GP-training programme are presented elsewhere.[30] Between November 2016 and March 2017, there were two trainer focus group discussions, and four trainee focus group discussions. Each focus group consisted of 6–11 participants, and lasted for 45–70 min. Demographic data (age, gender and length of career to date) on the participants were collected by means of a questionnaire.

Each focus group discussion was led by a skilled moderator (MRMV or NvD), while an assistant-moderator (LHAB or a research assistant) took notes.

The sessions started with a presentation on the purpose of the study, during which participants had the opportunity to ask questions. As a first step, we asked participants to explore their own definition of trust, and we encouraged participants to use their own definition of trust during the focus group discussion. In the next steps, an exploratory question initiated the discussion on trust between trainer and trainee, after which key questions were used to maintain the momentum. The topics covered (table 1) were drawn from the literature on trust in workplace-based medical education.[3 12 14 31] Data collection and data analysis were performed iteratively. The results of the data analysis were used to guide subsequent data collection.[26] The outcomes of the focus group discussions of both trainers and trainees were used to modify the topic list used in following focus group discussions with both trainers and trainees, making it possible to explore trainers' and trainees views and practical experience of trust in broader detail.

After each meeting, the moderator and assistant-moderator conducted a debriefing to discuss the course and the details of the discussion. The debriefing notes formed part of the dataset.

## Data analysis

Audio recordings of the discussions were transcribed verbatim and anonymised. In the first step, LHAB (who is a GP trainee) performed an inductive coding process in four focus group discussions. The first step in the coding process resulted in a code book with themes that emerged from the focus group discussions. The details of this code book were discussed at length with NvD (who is an MD and an educationalist), to ensure coding reliability. In the second step, LHAB and NvD performed an axial coding

| Table 1 | Topics used to facilitate focus group discussions | |
|---|---|
| **Opening question** | |
| What do you need to trust someone in the context of training? What factors promote or undermine trust? | |
| Key questions | |
| Trainers | Trainees |
| How does trust in a trainee evolve? | How do you recognise your trainer trusting you? |
| What factors are involved in the development of trust? | How did this trust relationship evolve? |
| How does the level of trust you show a trainee influence their work and learning in daily clinical practice? | In what ways did you foster the development of trust? |
| Is a trust relationship transferable to different individuals or clinical situations? | How does the level of trust shown to you by your trainer influence your work and learning in daily clinical practice? |
| | What factors would enable you to trust your trainer? |
| | How does the level of trust you show your trainer influence your work and learning in daily clinical practice? |

**Table 2** Participant characteristics (Amsterdam, 2017)

| | Trainers | Trainees |
|---|---|---|
| Age in years (median (range)) | 50.0 (39.0–62.0) | 29.0 (27.0–38.0) |
| Gender | | |
| Female (%) | 40.00 | 82.40 |
| Working experience in years | | |
| As GP trainer (median (range)) | 8.0 (3.0–22.0) | – |
| Before start GP training programme (median (range)) | – | 1.6 (0.0–4.5) |

GP, general practitioner.

process involving the creation of overarching categories in the code book.In the third step, LB encoded the remaining two focus groups, using the code book derived from the axial coding process. As no new themes emerged from the last two focus groups, we assumed that data saturation had been reached. Once again, the code book was discussed in detail with NvD. The fourth step in the process of data analysis involved selective coding of the data. NvD, MRMV (who is a psychologist) and LHAB used the axial-coded code book to identify any themes that were relevant to the research question and to identify any relationships between the relevant themes. This process was supported by the use of memos and diagrams.[28 32 33] In the case of uncertainties or disagreement between the researchers during the coding process, the verbatim transcripts and audio-recorded data were consulted.

We used MAXQDA2018 (VERBI Software (2017), Berlin, Germany) for data analysis.

The participants were fully informed about the purpose of the study, participation was entirely voluntary, and individuals were able to withdraw at any time without having to give a reason. Prior to the focus group discussions, participants signed an informed consent form. All data were collected, stored and processed anonymously. No rewards were offered for participation in the study.

### Patient and public involvement
There were no patients involved in this study.

### RESULTS
Fifteen GP trainers and 34 trainee GPs participated in the study. Participant characteristics are displayed in table 2. Of the participating trainees, 82.4% was female. Overall, the atmosphere during the focus groups was pleasant and supportive. Participants felt safe to share their experiences, even though some experiences led to emotional reactions.

Based on our data, we created a model that is based on the elements of the relationship between trainers and trainees that contribute to the development of a feeling of trust in trainers and trainees (figure 1). The development of trust involves multiple factors and stages, and also shows some interactive aspects. Trust development starts with the first impression, based on which both trainers and trainees develop a level of basic trust in one another. Further development of the trust relationship between trainers and trainees during the course of the training programme is related to the process of training the trainee. Trainers and trainees adopt different approaches to building trust and have different goals. Trainers base the trust in their trainee on the performance of the trainee in working and learning in daily clinical practice, while trainees focus on the personal relationship with

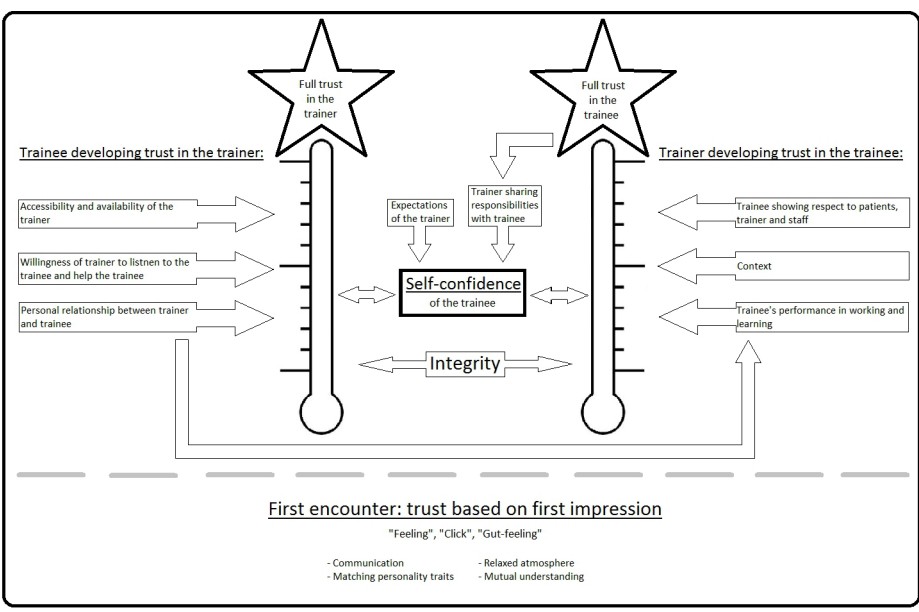

**Figure 1** Model for the mutual trust relationship between trainers and trainees.

 Bonnie LH.A, *et al. BMJ Open* 2020;**10**:e036593. doi:10.1136/bmjopen-2019-036593

their trainer. Trainee self-confidence is a major factor affecting the development of the trust relationship in both directions. Below we further describe the components of the development of trust.

## Stage 1: basic trust

Based on their first impressions, trainers and trainees each start to develop basic trust in the other. Basic trust is not typically related to the training of the trainee, but entails a more generic kind of trust in other people. Basic trust forms the foundation of the trust relationship between trainers and trainees. The process of developing basic trust seems difficult to put into words. Both trainers and trainees refer to it as a 'feeling', or 'click' that they experience with the other person. Trust antecedents for boosting the development of basic trust that were identified by both trainers and trainees are the way of communicating with each other and a relaxed atmosphere during the first encounter. Additionally, matching personality traits and a feeling of mutual understanding, also support the trust development process in this first short period.

Quote (trainee GP, 2nd year): The development of trust starts with a good first impression. But it's not good if there is no 'click'.

Both trainers and trainees indicate that first impressions have a strong influence on their trust in each other. Once established, the trust relationship tends to be difficult to adjust, also when factors occur that might influence the degree of trust in the other person.

Quote (GP trainer): There is very little scope for modifying a first impression. Your impression of the trainee turns out to be virtually immutable.

## Stage 2: trust development related to the training of the trainee

Once trainers and trainees start working together in daily clinical practice, the focus of the trust relationship shifts towards the process of training the trainee, based on experiences during working and learning in daily clinical practice. Trainers and trainees have different approaches to and different goals for building a trust relationship, but joint conditions for trust development are also present.

### The development of a trainer's trust in their trainee

The behavioural trust of trainers is to entrust trainees to see patients independently. Trust antecedents that influence the behavioural trust of trainers are factors such as the trainee's performance in working and learning, context-dependent factors and the degree of respect trainees show to patients, trainers and practice staff.

Trainee performance in working and learning is reflected in the trainee's transparency, their openness about how they perform their work, and their ability to learn from previous feedback. Trainers tend to focus on how trainees communicate about their positive and negative (learning) experiences in the workplace, on how they deal with trainer feedback, whether they know their own limitations, and ask for help at the right moments.

Quote (GP trainer): Trust is also based on the trainee's openness about discussing past cases with me. If all I hear from them are success stories, while my assistant or patients give me conflicting reports, that gets me thinking: 'Wait a moment…'

Quote (GP trainer): We do not blindly trust trainees, believing them to be capable of dealing with any and all situations. We trust them because we know that, when they find themselves in situations of which they have no previous experience, they will be able to deal with them. They are aware of their own limitations, and they know when to ask for help.

Context-dependent factors also influence whether trainers place sufficient trust in their trainees to entrust them with care for specific patients. These include the nature of the patient's problem, and (special) situations in which the patient is presented. Trust is put to the test in emergency consultations, during house calls or during shifts at the Out-of-Hours GP Centre, when there is a greater risk of serious illness.

Quote (GP trainer): 'Emergency care is typically a topic in which, as a trainer, I need to trust my trainee.'

Quote (GP trainer): 'In certain areas I think trainees can't do any harm, so I let them try to work independently.'

### The development of a trainee's trust in their trainer

Trainees have to trust their trainer in order to experience a safe learning environment. Trust antecedents involved in the development of a trainees' behavioural trust are based on the trainer's availability and accessibility, and on the trainer's ability to create an environment in which the trainee feels that the trainer is willing to listen to them and help them. If trainers are accessible, wherever and whenever required, this gives trainees a feeling of security and lowers the feeling of perceived risk, thereby enhancing their trust in their trainer.

Quote (trainee GP, 2nd year): I think it also has to do with accessibility. My GP trainer is very accessible, which increases my trust in him considerably.

Another trust antecedent influencing trainees' trust development is the personal interest that trainers show in their trainee. High personal interest creates a feeling of connectedness which, in turn, promotes the development of trust. Trainees find it more easy to be open about their performance and learning in daily clinical practice when they feel safe with, and connected to their trainer, especially in the case of problems and negative experiences.

Quote (trainee GP, 2nd year): I did not have a good relationship with my trainer, there was no real click at the personal level, which made it harder for us to

work together. It also made it harder for me to show my vulnerabilities…

The trainees noticed that, during short,non-GP,rotations (3–6 months), where several different trainers are involved in the training of the trainee, their personal relationship with their trainer was less intense than during year-long rotations in the GP practice, with a maximum of two trainers. During the short rotations, and in the case of multiple (more than two) trainers, trainees find it harder to develop a personal connection with and trust in their trainers.

Quote (trainee GP, 2nd year): In a whole year in training, you encounter all sorts of problems. It's good to discuss those problems with your trainer. If the individual relationship were less intense, then I would not discuss my personal problems. On short rotations I notice that there is much less of a personal connection.

### Joint conditions for trust development
We identified trust antecedents that influence the trust development process of both trainers and trainees.

#### Integrity
Both trainers and trainees build trust based on the integrity of the other person. Trainers and trainees assess each other's integrity in terms of consistent behaviour. They also compare each other's actions with their own work in daily practice and with their own moral standards. Consistent behaviour, and converging working styles and moral standards tend to boost their trust in each other.

Quote (GP trainer): Sometimes, I don't trust a given trainee, but I don't know why. Usually, it has something to do with their Professional role. Sometimes I think, 'I would not approach someone in that way, I do not want my patients to be treated like that.' Yet, the trainee has done nothing wrong in terms of their Medical Expert role.

Quote (trainee GP, 2nd year): I think, in fact, that it's easier to trust someone whose approach to work matches your own.

#### Trainee self-confidence
Trainee self-confidence is a major factor in the development of trust between trainers and trainees. If trainers entrust trainees to work independently in daily practice, this will improve trainee self-confidence. The effect is enhanced if trainers view trainees as equals and are willing to share their daily responsibilities with them.

If trainers and trainees have the same expectations of the trainee's performance, this boosts trainee self-confidence. However, if a trainee's degree of self-confidence deviates from their trainer's expectations this can impair the trainer's trust, as trainers feel that too much or too little self-confidence could impact patient safety. Such mismatches between the trainer's expectations and the trainee's actual

level of self-confidence can also impact the trainee's trust in their trainer, as in those cases trainees feel their trainer is not giving them sufficient support at the level where trainees feel they need it. As trainees are experiencing a lack of support from their trainer, this process will also negatively influence the self-confidence of the trainee.

Quote (GP trainer): Some trainees tend to overestimate their abilities, and your immediate reaction is that you don't want to entrust your patients to them.

Quote (trainee GP, 1st year): I get very uncomfortable if my trainer shows me a great deal of trust, because I wonder if I really deserve it.

### DISCUSSION
This study indicates that the process of mutual trust development between trainers and trainees in a workplace-based PGME-programme involves various stages, factors and interactive aspects. The process of mutual trust development starts at the first encounter, with the first impression forming basic trust. Within the formation of basic trust it is recognised that once a trust relationship is established, it is very hard to adjust this feeling of trust. This fits the model of Das and Teng, as it is recognised that behavioural trust again influences subjective trust.[25] Once trainers and trainees start working together in daily clinical practice, the focus of the trust development process shifts towards working and learning in the workplace. In this phase, trainers develop trust in their trainees in order to be able to entrust them with patient care. Trainees need to trust their trainer in order to experience a safe learning environment. Next to trainer-related and trainee-related factors, joint conditions for mutual trust development, like integrity and trainee self-confidence, are also involved in the mutual trust development process. The interplay between the joint conditions is an example of how behavioural trust of both trainer and trainee influences the subjective trust development process of themselves and the other person, leading to a mutual trust relationship. The findings of this study thereby support the ideas of Das and Teng that the trust development process is a looped and mutual process.[25]

When returning to the framework of a risk-based view of trust, we see that the opposite of subjective trust is perceived risk. When developing trust in the other person, subjective trust and perceived risk in the trainer–trainee relationship are both influenced by the competence (the ability to fulfil the task) and intentions (integrity and willingness to fulfil the task) of the other. This means that estimates of good competence and good intentions lead to high levels of subjective trust and low levels of perceived risk and the other way around.[25] Competence and intentions also play an important role in the mutual trust development process between trainers and trainees. Trainees evaluate a trainers competences regarding being an accessible trainer and the intentions of the trainer of being willing to help. Trainers on the other hand evaluate

the trainees competence of having insight in their own learning process and the intentions to be open about their working and learning process. Being aware of the role of competence and intentions in the trust development process, and in the differences in competence and intentions between trainers and trainees might help both trainers and trainees in the (more early) recognition of trust and risk. Perceived trust and perceived risk are dependent of personal factors,[3–5] and therefore, may also lead to different definitions for trust and risk between trainers and trainees. However, insight in the factors and processes of trust development might help both trainers and trainees not only to be able to discuss differences in the understanding of the mutual trust relationship with each other, but also to pay specific attention to these factors when developing a trust relationship or handling a trust issue.

It is recognised before that the development of a trust relationship between two people knows various stages. In the first stage, the involved persons have to get to know each other. Once they have become familiar with each other, the trust development process will increasingly be based on the knowledge about the other person and about the relationship that they have developed.[34] Tschannen-Moran and Hoy[35] assume that learners follow four stages in developing trust in their teachers: the initial impression phase, the exploration phase, the limit testing phase and the stability phase.[35 36] The findings of our study mainly reflect the initial impression stage (basic trust development) and the exploration and limit-testingstage (trust development focusing on working and learning in daily practice). In our study, the stage of development of basic trust (initial impression stage) seems to have a strong influence on further trust development process. This is in line with other studies saying that the amount of trust developed in the initial stage influences the likelihood to see factors and situations that further influence trust.[37]

In the second stage of trust development, when the trust development process focuses on working and learning in daily practice, trainers and trainees have different styles in trusting each other, and different goals for trusting each other. The factors influencing the trust of trainers in their trainees were the same ones identified by Hauer et al,[14] and Sagasser et al[15] in their studies into this process. Trainees in the second stage of trust development develop trust in their trainer based on the availability and accessibility of the trainer for the trainee, and on their personal relationship. This is in line with factors that Sutkin et al identified as students' criteria for good clinical teachers,[38] and with earlier findings that a caring attitude towards learners may support trust development from trainees in their trainer.[35 39] This might explain why trainees, more than trainers,value the personal relationship when developing trust in their trainer.

The trainees noticed that the trust relationship with their trainer was stronger during long rotations (a minimum of 1 year) than it was during short rotations (usually 3–6 months), especially due to the fact that they were able to develop a more intense personal relationship with their trainer. Hirsh et al found that this effect may be due to the more effective distribution of time during longitudinal rotations.[40] A more intense personal relationship during long rotations encourages trainees to discuss their performance with their trainer more often, leading to greater shared responsibility in identifying approaches to deal with any weaknesses in the trainee's performance. It also helped trainees to take a more constructive view of their trainer's feedback and to provide better patient care.[41] Longitudinal rotations may, therefore, not only lead to an improved learning environment with more learning opportunities for the trainee, it might also help to improve patient care. Other explanations for the difference between long and short rotations could be the differences in culture between working in hospitals and working in GP practices, as well as power differences and ethnocentric mismatches between GP trainees and their trainers in hospitals.

### Strengths and limitations
For this study, we were able to include a relatively large amount of trainers and trainees, in various stages of the training programme, which provided us with insights in many different experiences and viewpoints about the trust relationship between trainers and trainees.

Trust is a feeling, that might have a different definition for every single person.[4 5 42] This might cause challenges in investigating trust development. Although the feeling of trust might be very personal, participants in this study indicate the same factors involved in trust as reflected in available literature.[14 15 34–39] We, therefore, think that the results of this study are representative for the trust development process between trainers and trainees.

Of the participating trainees, 82.4% was female. Since 77% of trainee GPs in the Netherlands are female,[43] females were slightly over-represented in our study. We do not think that the results of our study are influenced by this variation and feel that the results are representative for Dutch trainee GPs.

This study is a single-centre study, focusing on the development of a trust relationship in the GP training programme. Therefore, the results might not be directly applicable in other PGME programmes. We were however able to identify factors that might influence the trust development process in (non-GP) short rotations or rotations with multiple trainers. We, therefore, think that aspects of our study may help various kinds of PGME programmes to further optimise the learning environment for their trainees.

### Future research
The trainees provided us with important insights concerning the effect of an educational relationship's duration on the trust relationship between trainers and trainees. However, we do not know which factors involved in short and long educational relationships influence the trust relationship between trainers and trainees.

Additionally, in this study, we did not explore the vision of trainers involved in short rotations. The distinction between long-term and short-term educational relationships and their effect on the trust relationships between trainers and trainees were therefore not fully explored and merit further study.

## Implications for practice

During the first impression, the foundation is made for the trust relationship between trainers and trainees. By introducing trainers and trainees to each other before the start of a training period, a solid foundation for the trust relationship can be established. When trainers and trainees are aware of the halo-effect of trust and of the differences in their needs concerning the trust relationship, the trust relationship and thereby learning in practice can be enhanced.

Furthermore, a long duration of a training relationship positively influences the trust relationship between trainers and trainees when compared with short-term training relationships, which might improve trainee performance and patient care.[41 44] When long-term training relationships are difficult to realise, the findings of this study might help trainers in short-term training relationships to improve the trust relationship between trainers and trainees, which could also improve the learning environment for trainees and patient care.

**Author affiliations**
[1]Department of General Practice/GP Speciality Training, Amsterdam UMC - Locatie AMC, Amsterdam, North Holland, The Netherlands
[2]Department of General Practice/GP Specialty Training, Leiden University, Leiden, South Holland, The Netherlands

**Acknowledgements** The authors wish to thank the participating trainers and trainees for their willingness to contribute to the study.

**Contributors** Author's contribution LHAB: Substantial contribution to the conception and design of the work, and to the acquisition, analysis and interpretation of data. Substantial contribution in drafting the work. Provides final approval of the version to be published. Agrees to be accountable for all aspects of the work. MRMV: Substantial contribution to the acquisition, analysis and interpretation of data. Substantial contribution in critically revising the work. Provides final approval of the version to be published. Agrees to be accountable for all aspects of the work. AWMK: Substantial contribution to the conception of the work and the interpretation of data. Substantial contribution in critically revising the work. Provides final approval of the version to be published. Agrees to be accountable for all aspects of the work. NvD:Substantial contribution to the conception and design of the work, and to the acquisition, analysis and interpretation of data. Substantial contribution in drafting and critically revising the work. Provides final approval of the version to be published. Agrees to be accountable for all aspects of the work.

**Funding** This publication was written as a part of the project 'The use of Entrustable Professional Activities in Assessment in General Practice Specialty Training' (project number 839130004), that has received fundings from the 'Netherlands Organisation for Health Research and Development' (ZonMW).

**Competing interests** None declared.

**Patient and public involvement** Patients and/or the public were not involved in the design, or conduct, or reporting, or dissemination plans of this research.

**Patient consent for publication** Not required.

**Ethics approval** This study was approved by the Ethical Review Board of the Netherlands Association for Medical Education (ERB-NVMO, file-number 664).

**Provenance and peer review** Not commissioned; externally peer reviewed.

**Data availability statement** Data are available on reasonable request.

**Open access** This is an open access article distributed in accordance with the Creative Commons Attribution 4.0 Unported (CC BY 4.0) license, which permits others to copy, redistribute, remix, transform and build upon this work for any purpose, provided the original work is properly cited, a link to the licence is given, and indication of whether changes were made. See: https://creativecommons.org/licenses/by/4.0/.

**ORCID iD**
Linda H.A. Bonnie http://orcid.org/0000-0002-5581-8761

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
