## [Reviewer comments · BMJ Open]

ARTICLE DETAILS

TITLE (PROVISIONAL)	Insight in the development of the mutual trust relationship between trainers and trainees in a workplace-based post-graduate medical training programme: a focus group study among trainers and trainees of the Dutch General Practice training programme.
AUTHORS	Bonnie, Linda; Visser, Mechteld; Kramer, Anneke; van Dijk, Nynke

VERSION 1 – REVIEW

REVIEWER	Allen F Shaughnessy Tufts University School of Medicine, Family Medicine
REVIEW RETURNED	16-Jan-2020

GENERAL COMMENTS	I enjoyed reading this paper describing how trust develops between general practice trainers and trainees. I have only a few comments that may serve to improve the paper. 1) I didn't see a definition of "trust" in the paper, a term that has psychological, sociological, and philosophical dimensions and definitions. I think a better term than "trust" is that you were exploring self-described "feeling of something I define as trust" rather than actual trust. In its broadest sense, "trust" is a construct in which one person feels he or she can rely on the actions of the other. Given the questions used in the focus groups, it seems the authors let each individual define what he or she meant by "trust", which may be fine. However, it is also problematic because what each person means by "trust", and therefore how it is achieved, is not well defined. As I step back and consider the results, I come to the conclusion that the model described in Figure 1 can serve as trust definitions, which are different for trainers and trainees. For trainers, it sounds like their implicit definitions revolved around their feeling that they could rely on both the reports of trainees of their past behavior (e.g., "I saw a patient with. . .") as well as their belief that the trainee could be relied upon to perform competently in future situations that the trainer could not control (entrustment). For trainees, on the other, hand, it seems the feelings of trust revolved more around their perception of the benevolence of the trainer (i.e., the psychological safety they would create) and the learning and patient care support the trainer would provide to the trainee at the time and in the future. For both groups, there is also the issue of power as it relates to trust that could be explored in the analysis.
--

	So, I suggest adding a description or descriptions of “trust” in the introduction and pointing out that you let participants use their own definitions of trust when answering your questions. I would like you to consider that from your focus groups you discovered what were the elements of the relationship that resulted in trainers’ and trainees’ development of a feeling of what they defined as trust. 2) I have an alternative explanation for trust relationships differing between long (years one and three) and short (year two) traineeships. It seems that an important variable, in addition to duration of the relationship, is that the supervisors in year two are not general practitioners. As a result, there may be power differentials, differences in cultures, and an ethnocentric mismatch between GP trainees and their specialist trainees that hamper the development of both trust and the feelings of trust on both sides (such is the reality, I think, in the United States). I suggest the authors consider whether they want to address this variable in their analysis. 3) I don’t see a methodological orientation in the paper. I see your method (thematic analysis) but I don’t see a theoretical framework guiding your research. A conceptual framework is necessary to understand the strengths and limitations of your approach (see: Medical Education 2009; 43: 312–319. doi:10.1111/j.1365-2923.2009.03295.x). 4) Your COREQ checklist doesn’t seem to line up with your paper, making it hard to verify.
--	---

REVIEWER	Malcolm Moore ANU College of Medicine and Health Sciences, Medical School
REVIEW RETURNED	04-Feb-2020

GENERAL COMMENTS	Although this study isn't breaking new ground, it contextualises the study of trust to the GP context and successfully aligns its outcomes with the existing literature. The findings about the difference in factors affecting developing trust between trainers and trainees are interesting and provide some useful insights for trainers. I don't think there are any significant revisions required. I note with interest that 82.4% of trainees are female - I suggest that this is mentioned in the results and a comment made about the representativeness of this sample.
---

VERSION 1 – AUTHOR RESPONSE

Reviewer 1

* 1) I didn’t see a definition of “trust” in the paper, a term that has psychological, sociological, and philosophical dimensions and definitions. I think a better term than “trust” is that you were exploring self-described “feeling of something I define as trust” rather than actual trust. In its broadest sense, “trust” is a construct in which one person feels he or she can rely on the actions of the other. Given the questions used in the focus groups, it seems the authors let each individual define what he or she

meant by “trust”, which may be fine. However, it is also problematic because what each person means by “trust”, and therefore how it is achieved, is not well defined.

As I step back and consider the results, I come to the conclusion that the model described in Figure 1 can serve as trust definitions, which are different for trainers and trainees.

For trainers, it sounds like their implicit definitions revolved around their feeling that they could rely on both the reports of trainees of their past behavior (e.g., “I saw a patient with. . .”) as well as their belief that the trainee could be relied upon to perform competently in future situations that the trainer could not control (entrustment). For trainees, on the other, hand, it seems the feelings of trust revolved more around their perception of the benevolence of the trainer (i.e., the psychological safety they would create) and the learning and patient care support the trainer would provide to the trainee at the time and in the future. For both groups, there is also the issue of power as it relates to trust that could be explored in the analysis.

So, I suggest adding a description or descriptions of “trust” in the introduction and pointing out that you let participants use their own definitions of trust when answering your questions. I would like you to consider that from your focus groups you discovered what were the elements of the relationship that resulted in trainers’ and trainees’ development of a feeling of what they defined as trust.

- Thank you for your thoughts on this part of the manuscript and for your recommendations. We now start the introduction of the manuscript with a definition and a discussion on the definition of “trust”. (Trust (noun): “Firm belief in the reliability, truth, or ability of someone or something”.(1) Entrust (verb): “To put one’s trust in a person, with regard to a particular task or responsibility.”(2) Trust is a complex expectation, not easy to summarize in a simple definition. The amount of trust someone has in another person is very much dependent from personal factors, such as thoughts and motivations, as well as it is dependent of the context in which the trust has to be awarded.(3-5) Additionally, trust is not an established fact, as it can be gradual and variable.(5) , (P: 7,L: 98-106))

We have also included information in the “Data collection”-section, stating that we encouraged participants to explore their definition of trust and use their own definition of trust during the focus group discussions. (“As a first step, we asked participants to explore their own definition of trust. Participants were encouraged to use their own definition of trust during the focus group discussion.” (P: 11, L: 209-212))

We have included information in the results-section to point out that it are elements of the relationship between the trainer and the trainee that contribute to the development of the feeling of trust. (“Based on our data we created a model that is based on the elements of the relationship between trainers and trainees that contribute to the development of a feeling of trust in trainers and trainees. (Figure 1).” (P: 14, L: 260-262))

At last we have included information in the strengths and limitations-section in the discussion in order to point out that trust-development is a very subjective process, and that the meaning of trust might vary between different persons. Although the feeling of trust is very personal, participants indicate the same factors involved in the trust development process, we therefore think that these results are representative for the trust development process between trainers and trainees. (“Trust is a feeling, that might have a different definition for every single person. (4-6) This might cause challenges in investigating trust development. Although the feeling of trust might be very personal, participants in this study indicate the same factors involved in trust as reflected in available literature.(7-14) We therefore think that the results of this study are representative for the trust development process between trainers and trainees.

“(P: 22, L: 459-463))

* 2) I have an alternative explanation for trust relationships differing between long (years one and three) and short (year two) traineeships. It seems that an important variable, in addition to duration of the relationship, is that the supervisors in year two are not general practitioners. As a result, there may be power differentials, differences in cultures, and an ethnocentric mismatch between GP trainees and their specialist trainees that hamper the development of both trust and the feelings of trust on both sides (such is the reality, I think, in the United States). I suggest the authors consider whether they want to address this variable in their analysis.

- Thank you for your considerations. We think that to some extent these power differentials, differences in cultures and the ethnocentric mismatch also hold for the Dutch GP-training program. However, Dutch trainee GPs also have to participate in nursing homes, which in the Netherlands is also a primary care facilitation with a similar educational structure, in which the power differentials, differences in cultures and the ethnocentric mismatch are not as striking as they are in hospital settings. Since trainees also mention these nursing home rotations as rotations in which they find it difficult to develop a personal relationship with their trainer, we believe that the power/cultural/ethnocentric differences between primary care and hospital care settings are not the primary reason for the differences in the development of the feeling of trust. We did however include a consideration in the discussion-section concerning this subject. (Other explanations for the difference between long and short rotations could be the differences in culture between working in hospitals and working in GP-practices, as well as power-differences and ethnocentric mismatches between GP-trainees and their trainers in hospitals. (P: 22, L: 449-452))

We do think that the finding of the differences between short and long traineeship rotations merits further evaluation. Our finding was that trainees indicate a difference in trust development between short and long rotations, future research could benefit from evaluating which factors involved in short and long rotations cause this difference in trust development. We have included a recommendation for future research in the discussion-section. (“Future research

The trainees provided us with important insights concerning the effect of an educational relationship’s duration on the trust relationship between trainers and trainees. However, we do not know which factors involved in short and long educational relationships influence the trust relationship between trainers and trainees. Additionally, in this study we did not explore the vision of trainers involved in short rotations. The distinction between long-term and short-term educational relationships and their effect on the trust relationships between trainers and trainees were therefore not fully explored and merit further study.”(P: 23, L: 477-483))

Additionally, literature on the effects of long rotations and longitudinal training relationships is increasing, suggesting that longitudinal training relationships have a positive influence on various aspects of trainee learning, especially due to the fact that trainees really get to engage in everyday working, together with their trainer.(3, 15, 16) Our hope is that in the future long educational relationships and exchange of trainees between primary and secondary/tertiary care will help us to overcome those differences in power, culture and ethnocentric mismatch.

* 3) I don’t see a methodological orientation in the paper. I see your method (thematic analysis) but I don’t see a theoretical framework guiding your research. A conceptual framework is necessary to understand the strengths and limitations of your approach (see: Medical Education 2009: 43: 312–319. doi:10.1111/j.1365-2923.2009.03295.x).

- Thank you very much for this methodological improvement of our manuscript. In our studies on this subject we use the risk-based view of trust (Das and Teng) as a guiding framework for data analysis. We did include some information in the methods-section. (For the leading conceptual framework we chose to adopt the risk-based view of trust (Das and Teng, 2004)(17), where we assume that trust

development is an ongoing process. We also assume that developing trust will have certain mutual aspects, because both parties involved in the trust development process have to take a risk in order to be open to each other's actions, and be able to develop trust in each other.(17) (P: 9, L: 162-166))

* 4) Your COREQ checklist doesn't seem to line up with your paper, making it hard to verify.

- Thank you for your attention, we have re-filled the form using the revised version of the manuscript without track changes. It might be possible that page- and line-numbers change with building the PDF-file for submitting the article. We hope that the renewed COREQ checklist now does line up with the paper.

Reviewer 2

* Although this study isn't breaking new ground, it contextualises the study of trust to the GP context and successfully aligns its outcomes with the existing literature. The findings about the difference in factors affecting developing trust between trainers and trainees are interesting and provide some useful insights for trainers. I don't think there are any significant revisions required. I note with interest that 82.4% of trainees are female - I suggest that this is mentioned in the results and a comment made about the representativeness of this sample.

- Thank you very much for your thorough review of the manuscript. Since 77% of the GP-trainees in the Netherlands is female, females are overrepresented in our study, we included some information about the study population in the results-section. Since the study population is relatively small (when compared to large cohort studies), we do not think that this difference affected the results of our study (our study required 2 more male trainees instead of female trainees) and that the results of our study are representative for the Dutch GP-training program. We did include some information in the discussion-section.(Of the participating trainees, 82.4% was female. Since 77% of trainee GPs in the Netherlands are female(18), females were slightly overrepresented in our study.. We do not think that the results of our study are influenced by this variation and feel that the results are representative for Dutch trainee GPs. (P: 22, L: 465-468))

Literature

1. Oxford Dictionaries [Definition for "Trust"]. Available from: <https://www.oed.com/view/Entry/207004?rskey=RegA4O&result=1#eid>.
2. Oxford Dictionaries [Definition for "Entrust"]. Available from: <https://www.oed.com/view/Entry/63012?redirectedFrom=entrust#eid>.
3. Hauer KE, Ten Cate O, Boscardin C, Irby DM, Iobst W, O'Sullivan PS. Understanding trust as an essential element of trainee supervision and learning in the workplace. *Adv Health Sci Educ Theory Pract.* 2014;19(3):435-56.
4. Burke CS, Ims DE, Lazzara EH, Salas E. Trust in leadership: A multi-level review and integration. *The Leadership Quarterly.* 2007;18:606-32.
5. Dietz G, Den Hartog DN. Measuring trust inside organisations. *Personnel Review.* 2006;35:557-88.
6. Hauer KE, Soni K, Cornett P, Kohlwes J, Hollander H, Ranji SR, et al. Developing entrustable professional activities as the basis for assessment of competence in an internal medicine residency: a feasibility study. *J Gen Intern Med.* 2013;28(8):1110-4.
7. Lewicki RJ, Bunker BB. Developing and maintaining trust in work relationships. In: Kramer R, Tyler T, editors. *Trust in organizations.* Thousand Oaks, CA: Sage; 1996. p. 114-39.
8. Tschannen-Moran M, Hoy WK. A multidisciplinary analysis of the nature, meaning, and measurement of trust. *Rev Educ Res.* 2000;70(4):547-93.
9. Tschannen-Moran M. *Trust and collaboration in urban elementary schools.* Columbus: Ohio State University; 1998.
10. Gabarro JJ. The development of trust, influence, and expectations. In: Athos AG, Gabarro JJ, editors. *Interpersonal behavior: Communication and understanding in relationships.* Englewood Cliffs, NJ: Prentice Hall; 1978. p. 290-303.

11. McKnight DH, Cummings LL, Chervany NL. Initial trust formation in new organizational relationships. *Acad Manage Rev.* 1998;23:473-93.
12. Sutkin G, Wagner E, Harris I, Schiffer R. What makes a good clinical teacher in medicine? A review of literature. *Academic medicine : journal of the Association of American Medical Colleges.* 2008;83:452-66.
13. Hauer KE, Oza SK, Kogan JR, Stankiewicz CA, Stenfors-Hayes T, Cate OT, et al. How clinical supervisors develop trust in their trainees: a qualitative study. *Med Educ.* 2015;49(8):783-95.
14. Sagasser MH, Fluit CRMG, van Weel C, van der Vleuten CPM, Kramer AWM. How entrustment is informed by holistic judgements across time in a family medicine residency program: an ethnographic nonparticipant observational study. *Academic medicine : journal of the Association of American Medical Colleges.* Epub ahead of print.
15. Hirsh DA, Holmboe ES, ten Cate O. Time to trust: longitudinal integrated clerkships and entrustable professional activities. *Academic medicine : journal of the Association of American Medical Colleges.* 2014;89(2):201-4.
16. Bernabeo EC, Holtman MC, Ginsburg S, Rosenbaum JR, Holmboe ES. Lost in transition: the experience and impact of frequent changes in the inpatient learning environment. *Academic medicine : journal of the Association of American Medical Colleges.* 2011;86(5):591-8.
17. Das TK, B-S. T. The risk-based view of trust: a conceptual framework. *Journal of Business and Psychology.* 2004;19(1):85-116.
18. Versteeg S, Batenburg R. Figures from the registration of General Practitioners, Survey 2017. (Available in Dutch: Cijfers uit de registratie van huisartsen, Peiling 2017). Nivel; 2019.

VERSION 2 – REVIEW

REVIEWER	Allen F. Shaughnessy Tufts University School of Medicine, USA
REVIEW RETURNED	20-Mar-2020

GENERAL COMMENTS	I think the addition of the definitions of trust and the use of the conceptual framework make this paper much better. I suggest two minor additions that will help guide the interpretation and to align it better with the existing literature: 1) Devote a paragraph (or three) or two to explaining Das and Teng's model, specifically on how trust develops in the context of risk (trust antecedents, subjective trust, behavioural trust). Your interpretation of their work, as it applies to yours, will be very helpful to readers. 2) In the discussion, return to Das and Teng and explain how trust and risk are intertwined (see Figure 2 of your reference 25) and how faculty members can use your results to engender trust and examine their ability to trust in their residents. This explanation could be somewhat extensive
--

REVIEWER	Malcolm Moore ANU College of Medicine and Health Sciences, Medical School
REVIEW RETURNED	22-Mar-2020

GENERAL COMMENTS	Thanks for answering my query re gender differential.
---

VERSION 2 – AUTHOR RESPONSE

Reviewer comments

Reviewer 1

Reviewer Name: Allen F. Shaughnessy

Institution and Country: Tufts University School of Medicine, USA Please state any competing interests or state 'None declared': None

Please leave your comments for the authors below I think the addition of the definitions of trust and the use of the conceptual framework make this paper much better. I suggest two minor additions that will help guide the interpretation and to align it better with the existing literature:

1) Devote a paragraph (or three) or two to explaining Das and Teng's model, specifically on how trust develops in the context of risk (trust antecedents, subjective trust, behavioural trust). Your interpretation of their work, as it applies to yours, will be very helpful to readers.

Thank you for this suggestion, the additional information provides more insight in how we interpreted the data. We included additional information about the interpretation of the risk-based view of trust in the "Study design"-section of the paper.

(As leading conceptual framework we chose to adopt a risk-based view of trust (Das and Teng, 2004).(1) By using this framework, we assume that trust and risk are theoretically each other's opposites. This means that when we trust someone, we experience a low risk for us in trusting the other person (perceived risk), and we are willing to accept that risk. On the other hand, when we do not trust someone, the perceived risk in trusting the other person is high, and we are not willing to accept that risk. The feeling of a certain amount of trust in someone is called subjective trust, the perception of trust. Subjective trust is then influenced by personal and situational factors, the so-called trust antecedents in the terminology of Das and Teng. The interplay between subjective trust and trust antecedents leads to certain behaviour in taking risks and trusting someone, also known as behavioural trust.(1)

Within the framework of a risk-based view of trust, the outcomes of behavioural trust may again inform and influence a person's subjective trust. Additionally, they believe that trusting somebody, and therefore taking a risk in that person, can boost the development of a mutual trust relationship.(1) We use these assumptions as the basis of our study, as we aim to explore how the mutual trust relationship between trainers and trainees develops, and which factors are involved in this process. (P9-10, L153-168))

Additionally, throughout the "Results"-section of the paper we identified trust antecedents, perceived trust, perceived risk and behavioral trust among our results. (P17-19, L299-425) This creates a better transition to the discussion section. Throughout the "Discussion"-section of the paper we made some adjustments in order to better align the discussion. (P23-28, L434-558)

2) In the discussion, return to Das and Teng and explain how trust and risk are intertwined (see Figure 2 of your reference 25) and how faculty members can use your results to engender trust and examine their ability to trust in their residents. This explanation could be somewhat extensive

Thank you for this suggestion, the additional information provides a link to the conceptual framework and adds an extra dimension to the discussion. We included additional information in the "Discussion"-section of the paper.

("When returning to the framework of a risk-based view of trust, we see that the opposite of subjective trust is perceived risk. When developing trust in the other person, subjective trust and perceived risk are both influenced by the competence (the ability to fulfil the task) and intentions (integrity and willingness to fulfil the task) of the person we have to trust. This means that estimates of good competence and good intentions lead to high levels of subjective trust and low levels of perceived risk. And the other way around.(1) Competence and intentions also play an important role in the mutual trust development process between trainers and trainees. Trainees evaluate a trainer's competences regarding being an accessible trainer and the intentions of the trainer of being accessible and willing to help. Trainers on the other hand evaluate the trainee's competence of having insight in their own learning process and the intentions to be open about their working and learning process. Being aware of the role of competence and intentions in the trust development process, and in the differences in competence and intentions between trainers and trainees might help both trainers and trainees in the (more early) recognition of trust and risk. Perceived trust and perceived risk are highly dependent of personal factors(2-4), leading to different definitions for trust and risk between trainers and trainees. However, insight in the factors and processes of trust development might help both trainers and trainees not only to be able to discuss differences in the understanding of the mutual trust relationship with each other, but also to pay specific attention to these factors when developing a trust relationship or handling a trust-issue." (P23-24, L445-463))

Reviewer 2

Reviewer Name: Malcolm Moore

Institution and Country:

Australian National University

Australia

Please state any competing interests or state 'None declared': None declared

Please leave your comments for the authors below Thanks for answering my query re gender differential.

You are welcome, we think it adds an extra dimension to the discussion.

Literature

1. Das TK, B-S. T. The risk-based view of trust: a conceptual framework. *Journal of Business and Psychology*. 2004;19(1):85-116.

2. Hauer KE, Ten Cate O, Boscardin C, Irby DM, Iobst W, O'Sullivan PS. Understanding trust as an essential element of trainee supervision and learning in the workplace. *Adv Health Sci Educ Theory Pract*. 2014;19(3):435-56.

3. Burke CS, Ims DE, Lazzara EH, Salas E. Trust in leadership: A multi-level review and integration. *The Leadership Quarterly*. 2007;18:606-32.

4. Dietz G, Den Hartog DN. Measuring trust inside organisations. *Personnel Review*. 2006;35:557-88.

VERSION 3 – REVIEW

REVIEWER	Tufts University School of Medicine, Family Medicine
REVIEW RETURNED	03-Apr-2020
GENERAL COMMENTS	Wow -- I really like the additions, which contextualize your findings within a theoretical model. I think this paper will be very valuable to educators interested in developing trust relationships!